# Effects of Short-Term Heat Stress on the Development, Reproduction, and Demographic Parameters of *Phytoseiulus persimilis* (Acari: Phytoseiidae)

**DOI:** 10.3390/insects16060596

**Published:** 2025-06-05

**Authors:** Hajar Pakyari, Rostislav Zemek

**Affiliations:** 1Department of Plant Protection, Tak.C., Islamic Azad University, Takestan P.O. Box 34819-49479, Iran; 2Institute of Entomology, Biology Centre CAS, 370 05 Ceske Budejovice, Czech Republic; rosta@entu.cas.cz

**Keywords:** predatory mites, extreme temperature, developmental time, population parameters, reproduction

## Abstract

This study investigated how short-term exposure to high temperatures affects *Phytoseiulus persimilis*, a predatory mite widely used to control spider mite pests in agriculture. As climate change leads to more frequent and intense heat waves, understanding how such heat stress impacts both pests and their natural enemies is crucial. The research exposed mite eggs and adults to high temperatures for four hours and monitored their development, survival, and reproduction. The results showed that, while warmer conditions sped up development, extremely high temperatures (42 °C) prevented eggs from hatching. Adult mites also lived shorter lives and produced fewer offspring at higher temperatures, with the best population growth observed at 36 °C. These findings suggest that rising temperatures could reduce the effectiveness of this beneficial predator, potentially threatening pest control in crops. By identifying the temperature limits and optimal conditions for *P. persimilis*, this research provides valuable insights for future pest management strategies. It helps farmers and agricultural planners prepare for climate challenges and maintain sustainable, chemical-free crop protection.

## 1. Introduction

Abiotic variables such as wind, humidity, and temperature have the most significant effects on biological control effectiveness. Numerous natural predators are commercially produced indoors, and hence must contend with environmental changes once they are released into the outdoors, often exceeding their typical range, which can cause injury or even death [1,2]. In particular, exposure to high temperatures can lead to altered metabolic rates, reduced fertility, and shortened lifespans, with a severe influence on population dynamics [3]. Consequently, a decrease in predation and survival, as well as the inability to increase the population, may potentially have an impact on biological control efficiency.

Organisms’ ability to withstand extreme temperature events, such as heat waves, will become increasingly important, as the severity, duration, and frequency of these events will continue to rise worldwide due to global warming [4,5,6,7]. Heat waves, defined as brief periods of stressfully high temperatures [8], usually happen in the summer when environmental temperatures generally fluctuate [9]. Brief heat stress can cause ecological and physiological harm to pests and their predators [10,11] and, in turn, have an impact on crop–pest dynamics [12] as well as on food pest–natural predator interactions [13]. Consequently, heat waves have large direct and indirect impacts on crop production [14]. Because climate models suggest the intensification of heat waves in the future, more frequent and severe occurrences of heat waves and mega heat waves are anticipated under the future warmer climate, and their distribution will vary significantly [5].

Temperature is a fundamental abiotic factor influencing the physiological processes, development, growth, and population dynamics of mites [9,15,16,17]. Optimal temperatures enhance mite survival; however, exceeding a certain threshold can lead to inhibited growth and mortality [18]. In natural environments, mites typically have sufficient time to acclimate to seasonal temperature variations. However, extreme temperatures can occur rapidly, giving mites limited time and resources to adapt to them and recover from the resultant damage [19]. Similarly to other arthropods, the consequences of heat stress on mites depend on the frequency, amplitude, and duration of the heat stress, as well as the feeding status, developmental stage, sex, and reproductive condition of the mites [20,21,22].

The impact of heat stress on mites has been extensively documented. Most studies revealed that heat stress adversely affects mite behavior, growth, reproduction, survival, and progeny fitness [9,20,23,24]. Additionally, heat stress causes harmful carryover effects on physiological traits across generations [25].

In agricultural systems, predatory mites such as *Phytoseiulus persimilis* Athias-Henriot play an important role in managing pest populations in many countries. Their advantage, among others, is that, as a specialist predator of spider mites, they can deal with dense spiderwebs [26]; they also have a high predation rate [27], making them an important biological control agent extensively used to suppress spiders, particularly the two-spotted spider mite *Tetranychus urticae* Koch, under field and greenhouse conditions [28,29]. The optimal temperature range for *P. persimilis* is between 17 and 28 °C [18]. Constant temperatures at or above 35 °C result in immature *P. persimilis* individuals not reaching adulthood [30], while, at 40 °C, adult females cease movement. The mean temperature at which heat coma is encountered is 41.1 °C [31]. However, although entry into heat coma effectively identifies the upper lethal temperature, the breeding of thermo-resistant lines of *P. persimilis* increased their resistance to temperatures of 40–42 °C by 8–11 times [32]. When this mite was exposed to extreme heat waves during its juvenile development, it developed faster, reached a smaller size at maturity [11], and consumed more prey [33]. Reproduction in *P. persimilis* increased under extreme heat waves but was not affected by juvenile acclimation [34]. Predator females also laid smaller eggs during extreme heat waves [35]. The effect of the short-term exposure of *P. persimilis* eggs and adults to high temperatures on the life table parameters of this predatory mite, to our knowledge, is currently unknown.

The present study, therefore, aimed to examine *P. persimilis*’ responses, namely its development, reproduction, and life table parameters, to short-term exposure to elevated temperatures, to comprehend its tolerance to a high-temperature environment. Knowing the effects of heat stress on various life stages of natural predators could yield essential insights for studies in natural predator ecology [20,21,36]. Furthermore, pest control via natural predators through introduction, conservation, or augmentation would be more efficacious if the heat resistance of biocontrol agents across different life stages were understood [37,38]. Moreover, the results can offer better directions for commercial predatory mite breeding implementation when releasing *P. persimilis* at high temperatures outdoors and/or indoors.

## 2. Materials and Methods

### 2.1. Rearing of Prey and Predator

Adults of the species *T. urticae* and *P. persimilis* were collected from cucumber and bean greenhouses in Pakdasht County in Tehran province. These were kept fresh on detached bean leaves that were put upside down on a layer of moist sponge in a 15 × 10 × 5 cm plastic container. Dampened tissue was placed around the sponge borders to provide moisture and keep mites from escaping. A 2 cm diameter hole was drilled into the container’s lid and covered with fine cloth mesh to provide ventilation. In a growth chamber, the containers were maintained at a 16:8 (L:D) photoperiod, 25 ± 0.5 °C, and 75 ± 5% relative humidity. *Phytoseiulus persimilis* were moved to a fresh container every second day. The predatory mite colony was maintained on *T. urticae* in this way for three months before using the mites in experiments.

### 2.2. Test Arena

*Phaseolus calcaratus* Roxburgh “Goli” seedlings were cultivated in greenhouses in pots 15 cm in diameter until they reached the fourth or fifth leaf stage. *Tetranychus urticae* were kept on detached bean leaves (3 cm in diameter) with thin veins. In a plastic dish measuring 1 cm deep and 6 cm in diameter, each leaf disk was placed upside down on a layer of moist sponge. To keep mites from escaping and to provide moisture, dampened tissue was placed around the border of the sponge. A 2 cm diameter hole in the middle of the container lid covered by a fine cloth mesh provided ventilation and prevented the mites from escaping.

### 2.3. Experimental Design

Thirty *T. urticae* eggs and a young gravid female *P. persimilis* were placed in the test arena. The female was removed 24 h later, leaving just one predator egg in the experimental arena. The eggs in the test arenas were then exposed to high temperatures of 36 °C, 38 °C, 40 °C, or 42 °C for four hours in a controlled environment chamber (Binder KBWS 240, Tuttlingen, Germany). The temperature range and relatively short exposure period were selected based on upper temperature limits reported in the literature [30,31,32] and because heat waves are more likely to occur during the middle of day, both in greenhouses and fields, for a single day or for a few days [8]. Previous studies on the effect of heat stress on phytoseiid mites, e.g., [20,23,24], used the same or similar temperatures and a duration of 2–6 h. Following exposure, all of the cells were stored and cared for in the environmentally controlled cabinet at 25 ± 0.5 °C, 75 ± 5% RH, and under a 16:8 (L:D) photoperiod. In the control group, the eggs were kept under the same conditions; i.e., they were not exposed to any heat stress. Fifty eggs were used in each heat stress treatment and fifty eggs were used in the control condition, and the growth and survival of *P. persimilis* individuals were monitored every 12 h. When mites reached adulthood, they were additionally exposed to 36 °C, 38 °C, and 40 °C for four hours. Males and females were then paired, provided with 75 prey eggs, and allowed to copulate in the same cell until the first egg was laid. When the females began to oviposit, the males were removed and the duration of the pre-oviposition period was recorded. For the heat stress-free control group, the same method was used. Survival rates and the number of eggs laid were recorded daily until all the females had died.

### 2.4. Statistical Analysis

The computer program TWOSEX MS-Chart Version 2020.06.16 available at https://lifetablechi.com/software/, accessed on 3 June 2025, was applied to analyze the collected data using the age-stage, two-sex life table theory [39]. The standard errors and variances of biological characteristics and population parameters were evaluated using the bootstrap resampling approach with 100,000 iterations [40]. The paired bootstrap test was used to assess treatment differences. The calculated population parameters, along with their definitions and equations, are listed in Table A1.

## 3. Results

The eggs exposed to 42 °C did not hatch. The development time for each stage of *P. persimilis* under different temperature conditions are listed in Table 1. At the egg stage, the development time decreased significantly with increasing temperatures. At 40 °C, the shortest development time was observed (1.39 ± 0.10 days), which was significantly different to that of the control group (1.83 ± 0.05 days). In the larva and protonymph stage, there were no significant differences in development time between different temperature conditions. The shortest development time for the deutonymph stage was seen at 40 °C (1.59 ± 0.08 days), with significant differences found between the control and 38 °C conditions (1.56 ± 0.07 days). Significant differences were observed in the length of the total pre-adult period between the control and the heat-stress treatments, particularly at 38 °C and 40 °C. The pre-adult survival rate was highest at 36 °C (0.70 ± 0.06) and lowest at 40 °C (0.62 ± 0.07). Both the female and male adult durations decreased significantly as the temperature increased, with the longest lifespan found in the control group and the shortest found under 40 °C conditions. Significant differences were observed across all treatments, with reductions becoming more pronounced at 38 °C and 40 °C.

Female and male longevity significantly differed across all treatments, with reductions becoming more pronounced at 38 °C and 40 °C (Table 2). Fecundity decreased significantly with an increase in temperature. The control group exhibited the highest fecundity (39 ± 1.39 eggs), while the lowest fecundity was found under 40 °C conditions (21.82 ± 0.75 eggs). The APOP remained consistent across most temperatures, with no significant differences between the control, 36 °C, and 38 °C conditions. The TPOP and the number of oviposition days were highest in the control group, while the lowest values were found at 40 °C. The percentage of females did not vary significantly across treatments, ranging from 0.44 ± 0.07 at 40 °C to 0.50 ± 0.07 at 36 °C (Table 2).

Comparing the population parameters of *P. persimilis* under different temperature conditions (Table 3) revealed that *r* and λ were highest at 36 °C, significantly higher than in the control. The control group exhibited the highest *R*_0_ value (18.72 ± 2.84 offspring/individual), while the lowest value was found at 40 °C (9.60 ± 1.55 offspring/individual), showing a drastic decline under severe heat stress. Significant differences were also observed in the mean generation time (*T*), with the duration becoming progressively shorter as the temperature increased (Table 3).

The *s_xj_* value for *P. persimilis*, which represents the likelihood that an individual would live to age *x* and stage *j*, allowed us to correctly determine stage overlap and distinction due to variations in development rates (Figure 1). The control group had the highest chance of a newly laid egg surviving to adulthood, with a probability of 0.51 for females and 0.21 for males. At 36 °C, the *s_xj_* value was also 0.51 and 0.21 for females and males, 0.49 for females and 0.21 for males at 38 °C, and 0.45 for females and 0.18 for males at 40 °C. Male and female individuals in the control group survived until days 28 and 18, respectively, while females and males survived until days 26 and 18 at 36 °C, days 25 and 17 at 38 °C, and days 23 and 17 at 40 °C (Figure 1).

Figure 2 shows the *l_x_*, *m_x_*, *l_x_m_x_,* and *f_xj_* values for different temperature conditions. Compared to the other temperature conditions, 36 °C produced the longest oviposition period and the highest *m_x_* value. The highest female age-stage-specific fecundity values for the control (day 23), 36 °C (day 18), 38 °C (day 18), and 40 °C (day 16) were 1.23, 1.63, 1.41, and 1.04 eggs, respectively. The *e_xj_* values of *P. persimilis*, which represent an individual’s expected remaining lifetime at age *x* and stage *j*, were predicted to differ across treatments. The control group had an age-stage life expectancy (*e_xj_*) of 6 days. For female adults at 36, 38, and 40 °C, the values were 5, 5, and 4 days, respectively (Figure 3). The maximum reproductive values (*v_xj_*), which reflect the individual’s potential for reproduction based on its current age and stage, as well as the probability of survival to reproductive maturity, were observed at ages of 10 days (13.16 d^−1^), 10 days (11.85 d^−1^), 9 days (10.61 d^−1^), and 8 days (8.79 d^−1^) under control, 36 °C, 38 °C, and 40 °C conditions, respectively (Figure 4).

## 4. Discussion

Global warming is expected to affect both pests [41] and biological control agents [42], as well as their interactions [43]. Consequently, climate change will result in new challenges for the pest management of various crops [44,45]. In this work, short-term heat stress, lasting for four hours, was administered to both eggs and adult females of the species *P. persimilis*, to determine the effect of heat stress on the development time, reproduction and population parameters of *P. persimilis*, thereby assessing how the species responds to high temperatures. We observed significant changes in these parameters, indicating that short-term heat stress had a negative impact on *P. persimilis*. Furthermore, similar impacts have been found in previous research [9,11,20,23,37,46].

A short period of heat stress is unlikely to cause direct mortality but may modify population dynamics via impacting life history traits, e.g., fertility [20]. The short-term exposure of the eggs of the predatory mite *Neoseiulus barkeri* (Acari: Phytoseiidae) to temperatures of 38 °C and 40 °C reduced its fitness [9,47], and the daily fecundity of female adults exposed to 40 °C was significantly lower that of female adults exposed to 38 °C [23]. In our investigation, predatory mites were able to develop and lay eggs properly after being exposed to temperatures of 36–40 °C for 4 h during the egg stage, although their development, oviposition, and longevity were significantly affected. On the other hand, *P. persimilis* eggs exposed to extreme temperatures of 42 °C did not hatch. This is in accordance with previous findings regarding *N. barkeri* eggs, which did not hatch when exposed to 42 °C for 2, 4, and 6 h [20,24]. However, when the newly emerged female and male *N. barkeri* adults were exposed to 42 °C for 4 h, the egg hatchability of the progeny generations was not affected; however, the females exhibited a markedly extended pre-oviposition period, a shortened oviposition period, and reduced fecundity and longevity [20]. It, thus, seems that phytoseiid eggs are more sensitive to heat stress than adults; nevertheless, more studies are required to confirm this hypothesis.

The results of the present paper showed the shortest development time of the juvenile stage when eggs were exposed to 40 °C (5.3 days), compared to the control group (6.5 days). Although this difference might seem to be modest, such changes can accumulate and impact population growth rates and synchrony with prey populations. However, the heat stress conditions in our experiments simultaneously reduced the longevity and fecundity of *P. persimilis*. When taken together, these opposing effects illustrate a life-history trade-off: faster development may not translate into higher fitness under stressful temperatures, which is demonstrated by the estimated values of an intrinsic rate of increase. This shows that predatory mites may choose between survival and reproduction in an adverse environment [9,23]. The fact that *P. persimilis* has the lowest daily fecundity and the shortest life span under unfavorable conditions might be attributed to energy being prioritized for reproduction. High temperatures can speed mite growth and eventually reduce adult individuals’ size [11]. According to Yuan et al. [48], damage induced by heat stress frequently comes at the expense of longevity and reproduction. It has been found that *Neoseiulus californicus* McGregor (Acari: Phytoseiidae) has a considerably reduced total fecundity at 30 °C than at 20 °C and 25 °C, with the lowest daily reproduction rates found at 20 °C. As the temperature increases, the pre-oviposition period, oviposition period, and overall longevity of adults decrease [49]. Adult females of *Mononychellus tanajoa* (Bondar) (Acari: Tetranychidae) did not deposit eggs after being exposed to 42 °C for 4 h [50]. The fertility of *N. barkeri* reduced by more than 50% after 2 h of exposure to 40 °C, demonstrating that high temperatures over a short duration could inhibit population increase [23]. Temperature has a greater impact on reproduction than on development and survival. The longer an adult female is exposed to high temperatures, the more likely their fecundity will be influenced [20,51]. The injury induced by high temperatures at the egg stage can be corrected later in its development, but the damage to reproduction is indirect. On the other hand, the damage caused by high temperature at the adult stage is more direct, resulting in a decrease in fecundity. High temperatures are necessary to limit sustained growth in a population. The acceleration of development and the reduction in reproduction might be attributed to long-term high-temperature adaptation for improved survival [23].

Our study was conducted at a constant temperature of 25 °C, except for the 4 h of exposure to heat stress. In real conditions, the temperature naturally goes up and down and such fluctuations have huge impacts on arthropods’ development, survival, and reproduction, as demonstrated by many studies. For example, parasitoids reared under fluctuating temperatures but at high average temperatures exhibited longer development times and reduced longevity compared to those reared under constant temperature regimes with corresponding means [52]. In *T. urticae*, the effect of temperature fluctuation depended on the range: egg-to-female adult development was faster under fluctuating temperatures from 12.5 to 27.5 °C than under constant temperatures, whereas the opposite trend was observed for temperatures above 30 °C [53]. Prey consumption by two types of phytoseiid mites and their oviposition rates were also affected by an alternating temperature regimen [54]. Therefore, temperature fluctuations should be considered in future heat stress experiments.

Most researchers found that relative humidity (RH) was the most critical factor influencing the egg hatching rate, functional responses, and reproduction of phytoseiid mites [55,56]. As a result, Walzer et al. [57] employed egg hatchability as a drought resistance indicator, effectively applied it to the screening of resistant strains, and conducted appropriate verification experiments. Nonetheless, the leaf boundary layer’s relative humidity in the canopy microclimate was greater than that of the surrounding air. In contrast, some researchers thought that some phytoseiid species were only marginally affected by relative humidity or an environment with low humidity [58]. Therefore, the primary factor influencing the field population and biological control efficacy of these phytoseiid mites was a high-temperature environment. On the other hand, *P. persimilis* was reported to be very sensitive to low humidity and to not hatch when the RH was below 55% [59]. Because humidity usually decreases when the temperature increases [34], it should also be taken into account when the effects of heat waves are studied, at least in the field, where, contrary to greenhouse conditions, humidity cannot be controlled. A short decrease in humidity during the day, however, does not necessary harm *P. persimilis*; the results of a study by Le Hesran et al. [60] suggested that *P. persimilis* eggs are able to adapt to variable humidity conditions and that even short periods of high humidity can mitigate the effects of drought.

High temperatures can also cause increased locomotory and dispersal abilities [61], faster respiration and metabolic rates [62], or even mortality [63]. According to Chen et al. [64] and Colinet et al. [65], they may also cause oxidative stress and increase the expression of heat shock proteins. While temperatures above 41 °C are lethal for adult females of *P. persimilis*, at 40 °C, females ceased movement but could recover [31]. It is, therefore, likely that even a short exposure to such extremely high temperatures could have negative effects on the foraging behavior of *P. persimilis*. Li et al. [24] reported that, with an increasing temperature, the attack rate of *N. barkeri* decreased, while its handling time increased, indicating the negative effects of short-term heat stress on the predation ability of a phytoseiid mite.

Heat stress will ultimately limit population expansion and continuation; this is because, while these physiological or behavioral responses may help the organism to survive the adverse environment, they also consume energy at the expense of the potential reproduction or survival rate [20,37,66]. In accordance with the findings of earlier research [11], we discovered that *P. persimilis* could grow, mate, and reproduce after short exposures to temperatures of 36, 38, and 40 °C, but its fitness was much lower than those exposed to 25 °C. This suggests that high temperature is a significant factor limiting sustainable population growth. The lower reproductive rate and faster development rate were most likely caused by a long-term adaptation mechanism to high temperatures in order to increase the probability of survival.

## 5. Conclusions

This study underscores the vulnerability of *P. persimilis* to heat stress, emphasizing its detrimental effects on development and demographic parameters. Addressing these challenges through targeted research and adaptive management is essential to sustaining the efficacy of *P. persimilis* in biological control programs, particularly in the context of global climate change. However, because environmental complexity, many plants, and other natural qualities cannot be fully replicated in small areas, most similar research, including ours, conducted in a laboratory setting might not be entirely representative of outdoor areas. Therefore, more research is required to study the effects of heat waves on predatory mites under greenhouse or field conditions.

Based on the results of the present study, we can recommend that both commercial transportation and field releases need to avoid high temperature exposure, even for short periods. In order to maximize *P. persimilis*’s effectiveness in bio-controlling *T. urticae* in both indoor and outdoor environments, growers must abstain from releasing the predatory mite in temperatures higher than 36 °C. Future research on the effects of heat stress on phytoseiid mites should include fluctuating temperatures, various humidity conditions, and the measurement of predation efficacy, e.g., functional response.

## Figures and Tables

**Figure 1 insects-16-00596-f001:**
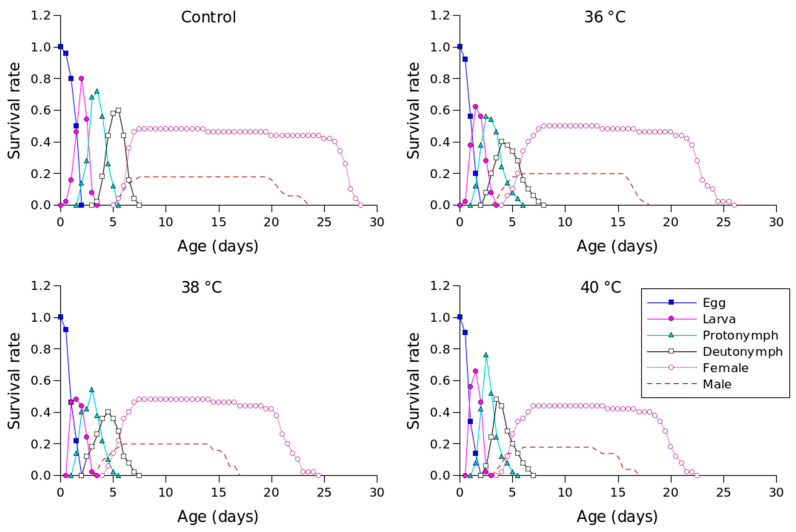
Age-stage-specific survival rates (*s_xj_*) of *Phytoseiulus persimilis* exposed to different heat stress conditions.

**Figure 2 insects-16-00596-f002:**
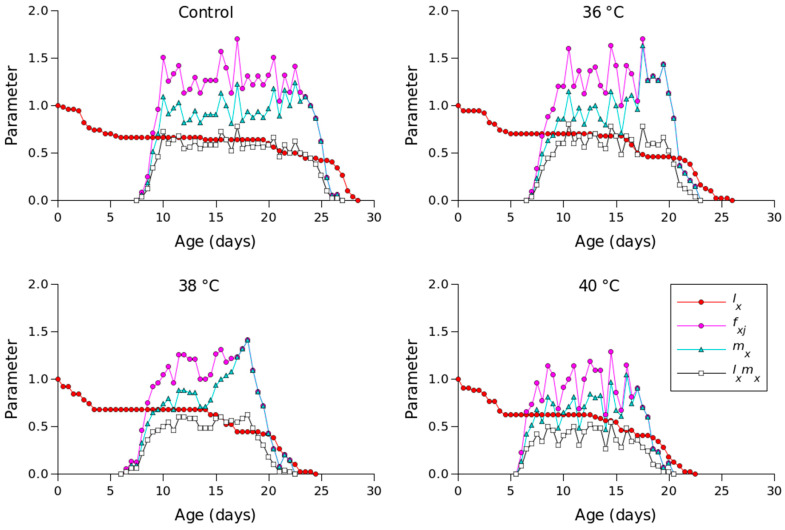
Age-specific survival rate (*l_x_*), age-specific fecundity (*m_x_*), age-specific maternity (*l_x_m_x_*), and age-stage-specific fecundity (*f_xj_*) of *Phytoseiulus persimilis* exposed to different heat stress conditions.

**Figure 3 insects-16-00596-f003:**
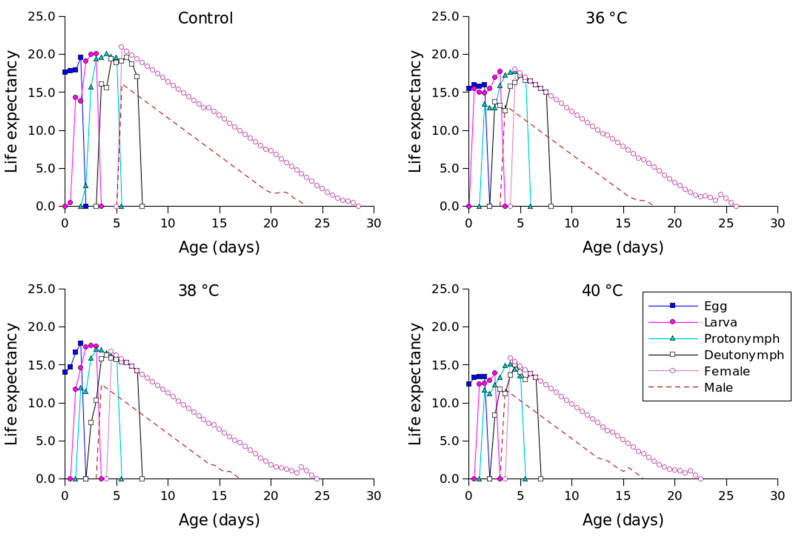
Age-stage-specific life expectancy (*e_xj_*) of *Phytoseiulus persimilis* exposed to different heat stress conditions.

**Figure 4 insects-16-00596-f004:**
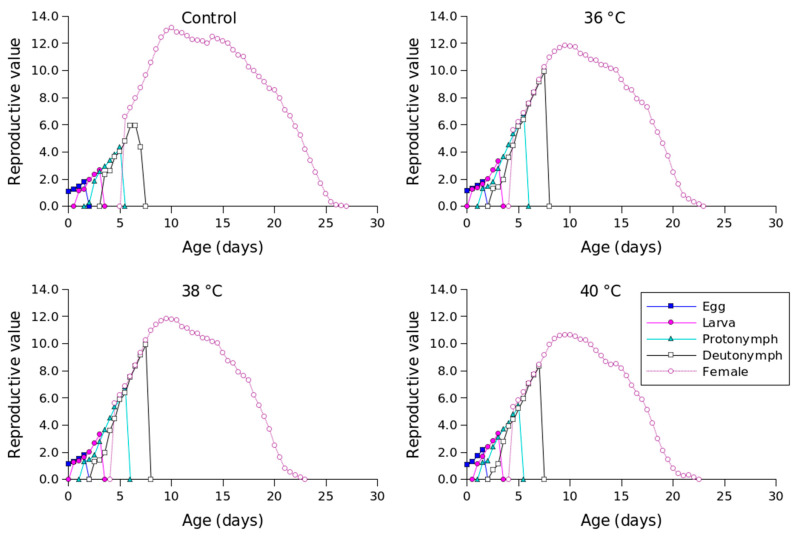
Age-stage-specific reproductive values (*v_xj_*) of *Phytoseiulus persimilis* exposed to different heat stress.

**Table 1 insects-16-00596-t001:** Mean (±SE) development time, survival rate, and adult duration of *Phytoseiulus persimilis* exposed to different heat stress.

Stage	Control	36 °C	38 °C	40 °C
Egg (days)	1.83 ± 0.05 a (49)	1.52 ± 0.09 bc (47)	1.71 ± 0.06 ab (46)	1.39 ± 0.10 c (45)
Larva (days)	1.1 ± 0.05 a (47)	1.08 ± 0.07 a (47)	0.93 ± 0.05 a (42)	0.86 ± 0.07 a (44)
Protonymph (days)	1.69 ± 0.09 a (37)	1.64 ± 0.10 a (40)	1.38 ± 0.08 a (39)	1.45 ± 0.13 a (39)
Deutonymph (days)	1.85 ± 0.05 a (33)	1.74 ± 0.08 ab (35)	1.56 ± 0.07 b (34)	1.59 ± 0.08 ab (31)
Pre-adult (days)	6.48 ± 0.10 a (33)	5.98 ± 0.19 ab (35)	5.79 ± 0.17 bc (34)	5.30 ± 0.18 c (31)
Pre-adult survival (proportion)	0.66 ± 0.06 b (33)	0.70 ± 0.06 a (35)	0.68 ± 0.07 ab (34)	0.62 ± 0.07 c (31)
Female adult (days)	19.92 ± 0.64 a (24)	16.50 ± 0.48 b (25)	15.46 ± 0.39 bc (24)	14.55 ± 0.38 c (22)
Male adult (days)	15.33 ± 0.29 a (9)	12.40 ± 0.16 b (10)	11.45 ± 0.23 c (10)	10.78 ± 0.28 c (9)

Values followed by different letters within the same row are significantly different (paired bootstrap test, *p* < 0.05).

**Table 2 insects-16-00596-t002:** Mean (±SE) longevity, fecundity, pre-oviposition periods, and oviposition days of *Phytoseiulus persimilis* exposed to different heat stress.

Biological Parameter	Control	36 °C	38 °C	40 °C
Female longevity (days)	26.40 ± 0.64 a (24)	22.48 ± 0.50 b (25)	21.25 ± 0.42 bc (24)	19.84 ± 0.38 c (22)
Male longevity (days)	21.61 ± 0.45 a (9)	16.80 ± 0.20 b (10)	15.95 ± 0.28 c (10)	15.28 ± 0.45 c (9)
Fecundity	39 ± 1.39 a (24)	32.08 ± 1.23 b (25)	25.96 ± 0.77 c (24)	21.82 ± 0.75 d (22)
APOP ^1^ (days)	2.65 ± 0.09 a (24)	2.70 ± 0.09 a (25)	2.75 ± 0.06 a (24)	2 ± 0.05 b (22)
TPOP ^2^ (days)	9.12 ± 0.12 a (24)	8.68 ± 0.19 b (25)	8.54 ± 0.18 b (24)	7.30 ± 0.19 c (22)
Oviposition days (days)	14.50 ± 0.55 a (24)	9.28 ± 0.33 b (25)	9.31 ± 0.25 b (24)	7 ± 0.22 c (22)
Sex ratio (proportion of females)	0.48 ± 0.07 a (33)	0.50 ± 0.07 a (35)	0.48 ± 0.07 a (34)	0.44 ± 0.07 a (31)

^1^ The period between the adult emergence and first oviposition. ^2^ The period from birth to first oviposition. Numbers in brackets indicate the number of individuals. Values followed by different letters within the same row are significantly different (paired bootstrap test, *p* < 0.05).

**Table 3 insects-16-00596-t003:** Mean (±SE) of population parameters of *Phytoseiulus persimilis* exposed to different heat stress.

Population Parameter ^1^	Control	36 °C	38 °C	40 °C
*r* (day^−1^)	0.191 ± 0.011 b	0.205 ± 0.012 a	0.188 ± 0.013 b	0.194 ± 0.016 b
*λ* (day^−1^)	1.211 ± 0.013 b	1.228 ± 0.015 a	1.206 ± 0.015 b	1.214 ± 0.019 b
*R*_0_ (offspring/individual)	18.72 ± 2.84 a	16.04 ± 2.33 b	12.46 ± 1.89 c	9.60 ± 1.55 d
*T* (days)	15.32 ± 0.20 a	13.54 ± 0.23 b	13.54 ± 0.23 c	11.68 ± 0.22 d

^1^ See Table A1 for definitions of the parameters. Values followed by different letters within the same row are significantly different (paired bootstrap test, *p* < 0.05).

## Data Availability

The raw data supporting the conclusions of this article will be made available by the authors on request.

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
