# Peer review of "Effects of Short-Term Heat Stress on the Development, Reproduction, and Demographic Parameters of Phytoseiulus persimilis (Acari: Phytoseiidae)"

_insects, 2025, doi:10.3390/insects16060596_

Round 1
Reviewer 1 Report
Comments and Suggestions for Authors
I have two suggestions for improving the manuscript:
-
I recommend using color figures to enhance clarity and visual impact.
-
It would be helpful to include the publication year when citing references within the text. For example, in Line 235, “Zhang et al.” should be revised to “Zhang et al. (2016).”

Author Response
Comment 1: I recommend using color figures to enhance clarity and visual impact.
Response 1: Figures were modified to color version and improved for beter clarity by enlarging fonts.
Comment 2: It would be helpful to include the publication year when citing references within the text. For example, in Line 235, “Zhang et al.” should be revised to “Zhang et al. (2016).”
Response 2: We did not modify the citation because MDPI style uses numerical format and adding year to author names would be confusing (it would look like mixture of both the numerical and the author-year formats).
Text in peer-review-46318224.v1.pdf file:
Comment 3: The manuscript titled -term Heat Stress on the Development, Reproduction and Demographic Parameters of Phytoseiulus persimilis (Acari: presents a timely and relevant study on the impact of transient high- temperature exposure on a key biocontrol agent. The research is well-conceived, and the application of age-stage, two-sex life table analysis provides a valuable quantitative framework. The findings have practical implications for the use of P. persimilis under climate-induced heat stress conditions. However, several aspects require clarification or improvement before the manuscript is suitable for publication.
Response 3: Thank you very much for a positive evaluation of the manuscript.
Comment 4: Sample Size
Each treatment involved only 30 individuals, which is relatively limited for life table studies. While the authors use robust statistical methods (bootstrap resampling), the small sample size may still affect the reliability of some parameters. It is recommended that the authors: Discuss this limitation explicitly in the Discussion section. Provide a brief justification for the chosen sample size (e.g., referencing similar studies or statistical power considerations). Consider reporting the number of replicates per treatment more clearly in Table 1 or in the Methods.
Response 4: As written in the middle of subsection 2.3. “Fifty eggs were used in each heat stress treatment and 250 eggs in control, …” the initial number was thus higher than 30 individuals (this was probably confused with 30 T. urticae eggs used as food). We believe that 50 individuals should be enough; for example in another life table study published recently in Insects (Borges, I.; Dury, G.J.; Soares, A.O. Population Growth Parameters of Scymnus nubilus Fed Single-Aphid Diets of Aphis fabae or Myzus persicae. Insects 2024, 15, 486. https://doi.org/10.3390/insects15070486) N=10 pairs of predators were used and N ranging from 10 to 38 were reported for biological trait parameters. Tables 1 and 2 were revised to include the number of individuals (N) used in calculation of each parameter.
Comment 5: Justification of Temperature Treatments
Response 5: Temperature range for heat shock treatments was selected based on upper temperature limits reported in literature which are more ellaborated in revised Introduction. The justification of conditions was also added into Material and Methods. The same or similar conditions of heat stress were used for example in the following studies:
Effects of heat stress on copulation, fecunditiy and longevity in newly-emerged adults of the predatory mite Neoseiulus barkeri (Acari: Phytoseiidae)
Guo Hao Zhang, Ya Ying Li, Kai Jun Zhang, Jin Jun Wang, Yi Qing Liu, Huai Liu
https://doi.org/10.11158/saa.21.3.5
The effects of short-term heat stress on functional response of Neoseiulus barkeri to Tetranychus urticae
Wei-Zhen Li, Tong Zhu, Hao-Long Li, Su-Qin Shang
https://doi.org/10.1111/jen.12954
Effects of short-term heat stress on the development and reproduction of predatory mite Neoseiulus barkeri (Acari, Phytoseiidae)
Wei-Zhen Li, Hao-Long Li, Zi-Kun Guo, Su-Qin Shang
https://doi.org/10.11158/saa.26.4.5
Comment 6: Consideration of Relative Humidity
Although relative humidity was controlled during the experiments, the potential interaction between high temperature and humidity is not discussed. Given its known influence on phytoseiid development and reproduction, a brief discussion of RH effects or limitations is warranted.
Response 6: Thank you for this important point. Some more text and references were added into Discussion.
Comment 7: Field Applicability
The authors mention the constraints of laboratory-based results. This point should be slightly expanded, perhaps with suggestions for future greenhouse or field-level validation studies.
Response 7: Conclusions were modified to emphasize the need for furure experiments.
Comment 8: Minor Revisions:
Figure Improvements: Font sizes in Figures 2 4 should be enlarged, and variable definitions (e.g., fxj, exj, vxj) included in the legends for clarity.
Response 8: Figures were modified to color version and improved for beter clarity (large fonts etc.).
Comment 9: Language Polishing: Minor grammatical issues and awkward phrasing should be corrected. A thorough language edit is recommended.
Response 9: The revised manuscript was edited by MDPI English editing service.
Comment 10: References: The references are generally appropriate, though the authors may wish to include additional citations on climate warming trends or IPM context to further support the relevance of the study. Additionally, some references appear to be redundant and should be removed to avoid unnecessary repetition and to improve the conciseness of the citation list.
Response 10: Additional references were included, e.g. at the beginning of Discussion, while redundant ones were removed.
Comment 11: Conclusion:
I think this manuscript addresses an important issue at the intersection of biological control and climate change. With the revisions suggested above, the manuscript would be significantly strengthened and would contribute meaningfully to the field of biological control under climate variability.
Response 11: Thank you for all suggestions and we hope that revised version can now be accepted for publication.
Reviewer 2 Report
Comments and Suggestions for Authors
The manuscript investigates the impact of short-term heat stress on the predatory mite Phytoseiulus persimilis, a biological control agent against spider mites in agriculture. The study focuses on the effects of exposing both eggs and adult mites to high temperatures (36°C, 38°C, 40°C, and 42°C) on their development, survival, reproduction, and population parameters. The research underscores the vulnerability of P. persimilis to heat stress and provides valuable insights for future pest management strategies.
The heat stress was applied for only four hours, which may not fully capture the cumulative effects of prolonged exposure to high temperatures in the field. Why was a four-hour duration chosen for the heat stress experiments?
What are the potential impacts of short-term heat stress on the behavior and foraging efficiency of P. persimilis?
First paragraph, it is better put a common name before a Latin name, eg. fruit fly Drosophila melanogaster.
Table 1, the number “1” in the “Control1” might be better removed.
The abbreviations (eg. APOP, TPOP) might be better explained under the corresponding table.
The title of Table 1, 2,3 should be more brief.
There is a mistake in the layout of the contents under Table 3.
The figures are not clear enough.
Figure 4 is confusing, what is reproductive value? How to understand “the reproductive value of the larvae”?
Author Response
Comment 1: The manuscript investigates the impact of short-term heat stress on the predatory mite Phytoseiulus persimilis, a biological control agent against spider mites in agriculture. The study focuses on the effects of exposing both eggs and adult mites to high temperatures (36°C, 38°C, 40°C, and 42°C) on their development, survival, reproduction, and population parameters. The research underscores the vulnerability of P. persimilis to heat stress and provides valuable insights for future pest management strategies.
The heat stress was applied for only four hours, which may not fully capture the cumulative effects of prolonged exposure to high temperatures in the field. Why was a four-hour duration chosen for the heat stress experiments?
Response 1: We expect that heat waves duration is usually short and occurre during middle of day both in greenhouse and the field (Robinson, 2001). Some previous studies also used similarly short exposure to heat stress. Explanation was added into the Material and Methods.
Comment 2: What are the potential impacts of short-term heat stress on the behavior and foraging efficiency of P. persimilis?
Response 2: Although the foraging behavior was not the objective of our study it is certainly a good question. We added this point into Discussion and refered to some previous studies.
Comment 3: First paragraph, it is better put a common name before a Latin name, eg. fruit fly Drosophila melanogaster.
Response 3: This part was rewritten as suggested by another reviewer. Many mite species, however, do not have common names.
Comment 4: Table 1, the number “1” in the “Control1” might be better removed.
Response 4: “1” was removed.
Comment 5: The abbreviations (eg. APOP, TPOP) might be better explained under the corresponding table.
Response 5: Definitions of abbreviations were added under table.
Comment 6: The title of Table 1, 2,3 should be more brief.
Response 6: All table and figure captions were shortened by deleting less important information.
Comment 7: There is a mistake in the layout of the contents under Table 3.
Response 7: Formating was corrected.
Comment 8: The figures are not clear enough.
Response 8: Figures were modified to color version and improved for beter clarity (large fonts etc.).
Comment 9: Figure 4 is confusing, what is reproductive value? How to understand “the reproductive value of the larvae”?
Response 9: The term reproductive value (vâ‚“j) refers to the expected future contribution of an individual at a specific age (x) and developmental stage (j) to the future population. It reflects the individual’s potential for reproduction based on its current age and stage, as well as the survival probability to reach reproductive maturity. For immature stages such as larvae, the reproductive value is not an actual reproductive output, but rather a projection of their expected future reproductive contribution if they survive to adulthood. A larva with a high reproductive value is one that has a high probability of surviving to adulthood and reproducing successfully. Explanation was added into Results and Table S1.
Reviewer 3 Report
Comments and Suggestions for Authors
This paper deals with high temperature effects on P. persimilis. Relevant topic, paper was a pleasure to read. Although I am not a big fan of the TWO SEX MS chart, the conclusions are relevant to the area of biocontrol.
General remarks
- No statistical values because of TWO SEX MS chart? Remains strange to publish a paper without any statistical values to back up the results.
Small remarks
- P2, L53-56: Why list these natural enemies? These are not the most known species (swirskii, Macrolophus, Orius, …)
- P2, L70: Suggest to add “directly and indirectly”
- P3, L137: it is not entirely clear to me why 250 control eggs were necessary and only 50 per heat treatment
- P4, L153: no capital “D” in development
- P4, L154: split development differently
- P8, LL231: parameters
- P8, L232: suggest to use another wording for “drop”. E.g. impact
- P8, L238: requires
- P8, L239: what do the authors mean with eliminate? Are being killed?
- P8, L 247: unver?
- P9, L286: optimally instead of steadily?
Author Response
Comment 1: This paper deals with high temperature effects on P. persimilis. Relevant topic, paper was a pleasure to read. Although I am not a big fan of the TWO SEX MS chart, the conclusions are relevant to the area of biocontrol.
Response 1: We sincerely thank the reviewer for the positive feedback.
Comment 2: General remarks - No statistical values because of TWO SEX MS chart? Remains strange to publish a paper without any statistical values to back up the results.
Response 2: Thank you veru much for highlighting an important point regarding the presentation of statistical results.
While traditional analyses often rely on ANOVA and associated statistics (e.g., F-values, p-values), the TWOSEX-MSChart method is based on non-parametric bootstrap resampling, which provides standard errors, confidence intervals, and significance tests specifically adapted for life table data that include variation across age, stage, and sex.
Significance between treatments in our tables is indicated by different letters, which are based on paired bootstrap tests with 100,000 iterations—a robust method for life table comparisons. Although these results are not expressed as traditional p-values, they provide a statistically valid basis for determining significant differences.
Comment 3: P2, L53-56: Why list these natural enemies? These are not the most known species (swirskii, Macrolophus, Orius, …)
Response 3: Yes, that is true and we therefore removed this part of text with named species.
Comment 4: P2, L70: Suggest to add “directly and indirectly”
Response 4: Added.
Comment 5: P3, L137: it is not entirely clear to me why 250 control eggs were necessary and only 50 per heat treatment
Response 5: That was a mistake, there were 50 eggs in the control, too so we corrected it.
Comment 6: P4, L153: no capital “D” in development
Response 6: Corrected
Comment 7: P4, L154: split development differently
Response 7: Spliting of “development” was done by MS-Word automatically, perhaps it could be corrected in final copy-editing.
Comment 8: P8, LL231: parameters
Response 8: Corrected.
Comment 9: P8, L232: suggest to use another wording for “drop”. E.g. impact
Response 9: “apparent drop” was replaced with “significant changes”
Comment 10: P8, L238: requires
Response 10: “require” is correct as it precedes with “will” (i.e. future tens).
Comment 11: P8, L239: what do the authors mean with eliminate? Are being killed?
Response 11: This sentence was rewritten.
Comment 12: P8, L 247: unver?
Response 12: Corrected to “under”
Comment 13: P9, L286: optimally instead of steadily?
Response 13: The sentence was rewritten as: "This suggests that the high temperature is a significant factor limiting sustainable population growth."
Reviewer 4 Report
Comments and Suggestions for Authors
The study by Pakyari and Zemek is generally well-written, with a solid design, clear structure, and solid analysis. It’s a good fit for this journal. That said, there are a few areas where I think some extra discussion and clarification would make it even stronger before it’s ready for acceptance.
Starting with Table 1, but also throughout the manuscript, I’m not entirely convinced about the practical significance of the differences shown, like 1.83 days versus 1.52 versus 1.39 days. Sure, the statistics say these are significantly different, but does that actually matter biologically? Would this small difference in development time really impact natural populations or biological control in any meaningful way? A bit more discussion on this could help readers understand what these numbers actually mean.
I’m also a bit confused by the statistical outputs in Table 1, and other tables, but focusing on Table 1 for now. The pre-adult survival rates are shown as 0.66 ± 0.06 (labeled 'b') and 0.70 ± 0.06 (labeled 'a'). Yet even though the error bars overlap, the results are still marked as significantly different. Normally, overlapping error bars suggest no significant difference, so I’m not sure how this is possible here. A quick explanation in the text would really help clear this up.
A bigger issue is the discussion section, which I think could be expanded significantly. One major thing missing is any talk about temperature fluctuations. Your study used constant temperatures, but in the real world, temperatures naturally go up and down. There’s a ton of research showing that these fluctuations can have huge impacts on insect development, survival, and reproduction. Constant temperatures just don’t capture that. It would be really helpful to discuss this, maybe mention some of the relevant studies on fluctuating temperature effects on insects, like the ones on parasitoid development and fecundity (https://doi.org/10.1093/jee/toz067; https://doi.org/10.1093/jee/toy429). Comparing your findings to those from studies using fluctuating temperatures would really help put your results in context.
I also think it’s important to explain why you used a four-hour heat stress period. Is there a reason this duration was chosen? Does it reflect real-world conditions insects would experience in the field? Just a quick note explaining this would give readers a better understanding of your setup. Again, a connection to the above-mentioned studies might help here, especially if those studies used different durations or fluctuating regimes. Highlighting how your four-hour exposure compares to those other conditions could clarify the relevance of your approach and the biological meaning of your findings.
If you can address these points, I think the paper will be a lot clearer and more impactful. Looking forward to seeing how it turns out. Thank you.
Author Response
Comment 1: The study by Pakyari and Zemek is generally well-written, with a solid design, clear structure, and solid analysis. It’s a good fit for this journal. That said, there are a few areas where I think some extra discussion and clarification would make it even stronger before it’s ready for acceptance.
Response 1: Thank you very much for a positive evaluation of the manuscript.
Comment 2: Starting with Table 1, but also throughout the manuscript, I’m not entirely convinced about the practical significance of the differences shown, like 1.83 days versus 1.52 versus 1.39 days. Sure, the statistics say these are significantly different, but does that actually matter biologically? Would this small difference in development time really impact natural populations or biological control in any meaningful way? A bit more discussion on this could help readers understand what these numbers actually mean.
Response 2: While a reduction from 1.83 to 1.39 days in egg development may seem modest, such changes can accumulate and impact population growth rates and synchrony with prey populations in applied biological control. In commercial mass-rearing systems or under rapid population expansion in the field, even small shifts in developmental timing can result in more generations per season or better timing of predator-prey interactions. Moreover, the shorter development times occurred under heat stress conditions, which simultaneously reduced longevity and fecundity. When taken together, these opposing effects illustrate a life-history trade-off: faster development may not translate into higher fitness under stressful temperatures.
Comment 3: I’m also a bit confused by the statistical outputs in Table 1, and other tables, but focusing on Table 1 for now. The pre-adult survival rates are shown as 0.66 ± 0.06 (labeled 'b') and 0.70 ± 0.06 (labeled 'a'). Yet even though the error bars overlap, the results are still marked as significantly different. Normally, overlapping error bars suggest no significant difference, so I’m not sure how this is possible here. A quick explanation in the text would really help clear this up.
Response 3: We agree that overlapping error bars can be confusing in interpreting significance. However, statistical comparisons in our study were conducted using the paired bootstrap test, which evaluates differences between treatment means based on resampling methods rather than overlap of standard errors or confidence intervals.This approach can yield significant differences even when standard errors appear to overlap because it accounts for the entire distribution of the data rather than just variability around the means.
Comment 4: A bigger issue is the discussion section, which I think could be expanded significantly. One major thing missing is any talk about temperature fluctuations. Your study used constant temperatures, but in the real world, temperatures naturally go up and down. There’s a ton of research showing that these fluctuations can have huge impacts on insect development, survival, and reproduction. Constant temperatures just don’t capture that. It would be really helpful to discuss this, maybe mention some of the relevant studies on fluctuating temperature effects on insects, like the ones on parasitoid development and fecundity (https://doi.org/10.1093/jee/toz067; https://doi.org/10.1093/jee/toy429). Comparing your findings to those from studies using fluctuating temperatures would really help put your results in context.
Response 4: This suggestion is very appreciated. A paragraph on this issue was added into Discussion with some references as examples of temperature fluctuation importance.
Comment 5: I also think it’s important to explain why you used a four-hour heat stress period. Is there a reason this duration was chosen? Does it reflect real-world conditions insects would experience in the field? Just a quick note explaining this would give readers a better understanding of your setup. Again, a connection to the above-mentioned studies might help here, especially if those studies used different durations or fluctuating regimes. Highlighting how your four-hour exposure compares to those other conditions could clarify the relevance of your approach and the biological meaning of your findings.
Response 5: We expect that heat waves duration is usually short and occurre during middle of day both in greenhouse and the field (Robinson 2001). Explanation was added into the Material and Methods. The same or similar conditions of heat stress were used for example in the following studies:
Effects of heat stress on copulation, fecunditiy and longevity in newly-emerged adults of the predatory mite Neoseiulus barkeri (Acari: Phytoseiidae)
Guo Hao Zhang, Ya Ying Li, Kai Jun Zhang, Jin Jun Wang, Yi Qing Liu, Huai Liu
https://doi.org/10.11158/saa.21.3.5
The effects of short-term heat stress on functional response of Neoseiulus barkeri to Tetranychus urticae
Wei-Zhen Li, Tong Zhu, Hao-Long Li, Su-Qin Shang
https://doi.org/10.1111/jen.12954
Effects of short-term heat stress on the development and reproduction of predatory mite Neoseiulus barkeri (Acari, Phytoseiidae)
Wei-Zhen Li, Hao-Long Li, Zi-Kun Guo, Su-Qin Shang
https://doi.org/10.11158/saa.26.4.5
Comment 6: If you can address these points, I think the paper will be a lot clearer and more impactful. Looking forward to seeing how it turns out. Thank you.
Response 6: Thank you for all suggestions and we hope that revised version can now be accepted for publication
Round 2
Reviewer 2 Report
Comments and Suggestions for Authors
The authors have revised the Ms according to the comments
Reviewer 4 Report
Comments and Suggestions for Authors
I don’t have any further comments. Everything looks good to me now. The authors have addressed all of my original concerns. Thanks again!